# Stock market forecasting research based on GA-WOA-LSTM

**Wu Huiyong, Zunlong Wang** [ID]*

School of Science, Shenyang University of Chemical Technology, Shenyang, Liaoning, China

* wangzunlong020714@163.com

## Abstract

With the increasing complexity and prosperity of global financial markets, stock market forecasting plays a critical role in investment decision-making, market regulation, and economic planning. This study proposes a hybrid prediction model that integrates Genetic Algorithm (GA), Whale Optimization Algorithm (WOA), and Long Short-Term Memory (LSTM) neural networks, referred to as the GA-WOA-LSTM model. In this framework, GA is employed to generate the initial population and perform global search for LSTM hyperparameter optimization, while WOA is applied to conduct local refinement of the search space. The LSTM model, known for its superior ability to capture nonlinear dependencies and long-term patterns in time series, is used to model and forecast future stock closing prices. The performance of the proposed model is evaluated on both training and test datasets using key metrics including Mean Absolute Error (MAE), Mean Absolute Percentage Error (MAPE), Root Mean Squared Error (RMSE), and the coefficient of determination ($R^2$). Experimental results demonstrate that the GA-WOA-LSTM model significantly outperforms traditional baseline models in terms of predictive accuracy and generalization capability. This research offers a robust and effective modeling strategy for financial time series forecasting and provides valuable insights for real-world financial applications.

## 1. Introduction

As one of the key financial markets in modern economies, the stock market's price fluctuations are influenced not only by economic factors but also by complex external factors such as political events and market sentiment. Therefore, the fluctuation of the stock market price is non-linear and non-stationary. How to effectively predict the stock price is of great significance to the financial and quantitative investment fields.

From an economic perspective, investors typically rely on traditional fundamental and technical analysis methods for stock market forecasting. However, these traditional strategies are overly theoretical and fail to effectively capture the complex relationships between time-series data. For example,BOLLERSLEV's [1] Generalized

**Data availability statement:** The data utilized in this study are publicly available on the Yahoo Finance platform (https://finance.yahoo.com/). All relevant datasets are freely accessible, ensuring the reproducibility of this research.

**Funding:** This work was supported by the Basic Scientific Research Project of the Department of Education of Liaoning Province (Project title: Generative AI in Multimodal Risk Scenario Simulation and Dynamic Emergency Decision-Making, Grant No. LJ212510149011 to HW and ZW). The funder had no role in study design, data collection and analysis, decision to publish, or preparation of the manuscript.

**Competing interests:** The authors have declared that no competing interests exist.

AutoRegressive Conditional Heteroskedasticity (GARCH) model and BOX's [2] Autoregressive Integrated Moving Average (ARIMA) model perform well in handling linear correlations in stock prices but struggle with non-linear relationships.

With the advancement of artificial intelligence and machine learning technologies, more machine learning algorithms and deep learning methods have been applied to stock market data prediction. Among these, Recurrent Neural Networks (RNN) and Convolutional Neural Networks (CNN) are core models in deep learning and represent key breakthroughs in the development of deep learning. Long Short-Term Memory (LSTM), a variant of RNN, addresses the issues of gradient explosion and vanishing gradients in long sequences [3]. Zeng Lifang et al. [4] Used BP neural network and Qi Xiaona et al. [5] used short-term memory network (LSTM) to predict the stock price, respectively, for stock price prediction, demonstrating better handling of non-linearity and non-stationarity in stock prices and achieving high predictive accuracy. Liu Zhen et al. [6] utilized an improved LSTM model based on Recurrent Neural Networks (RNN) for stock price forecasting.

In the field of optimization algorithms, the Whale Optimization Algorithm (WOA) is an innovative intelligent optimization method proposed by Seyedali Mirjalili and his team [7] in 2016, inspired by the predatory behavior of whales. Compared to other intelligent optimization algorithms, WOA requires fewer parameter adjustments, has a more stable search process, and is relatively easy to implement. These advantages make WOA uniquely competitive in the field of intelligent optimization. However, WOA still faces some challenges, such as lower solution accuracy and slower convergence speed. As a result, many researchers have made improvements to address these shortcomings. For instance, Kaur et al. [8] introduced chaotic mapping into the algorithm to adjust key parameters, providing an effective solution to the issue of unbalanced search and exploration. Chen et al. [9] combined Lévy flight with chaotic local search strategies, enhancing the optimization performance of WOA in complex environments.

In response to the challenges of strong nonlinearity, high volatility, and parameter sensitivity in financial market forecasting, traditional machine learning methods often fall short when dealing with complex time series data. These methods typically rely on linear assumptions, lack efficient hyperparameter tuning mechanisms, and are vulnerable to noise and outliers, resulting in unstable performance and limited prediction accuracy [10–12]. To address these limitations, this paper proposes an innovative hybrid forecasting model that integrates the Genetic Algorithm (GA), Whale Optimization Algorithm (WOA), and Long Short-Term Memory (LSTM) networks. The proposed model leverages the complementary strengths of intelligent optimization algorithms and deep learning. Specifically, GA is employed for its powerful global search capability to efficiently explore optimal combinations of LSTM hyperparameters, while WOA is utilized for its strong local refinement ability to fine-tune the parameters further, enhancing the model's adaptability and robustness in dynamic financial environments. The coordinated integration of GA and WOA effectively balances global exploration and local exploitation, overcoming the tendency of single algorithms to fall into local optima. By introducing this novel GA-WOA-LSTM architecture, the model

significantly improves its capacity to capture nonlinear and dynamic patterns in financial time series. Experimental results demonstrate that the proposed method achieves superior prediction accuracy and stability in stock market forecasting tasks, offering a promising approach for intelligent modeling of complex financial data.

## 2. Theoretical foundation and models

### 2.1. Genetic Algorithm (GA)

The Genetic Algorithm (GA) is an optimization technique that simulates the process of biological evolution, belonging to the class of evolutionary algorithms. It optimizes solutions by mimicking mechanisms such as natural selection, genetic inheritance, and mutation, thereby iteratively improving the solution to converge toward the optimal one. The fundamental steps of GA include population initialization, fitness evaluation, selection, crossover, mutation, and replacement [13–16].

Initially, GA starts with a randomly generated population, where each individual represents a potential solution. The algorithm then evaluates the fitness of each individual using a fitness function, selecting individuals with higher fitness values to serve as parents. These parent individuals undergo a crossover operation to generate offspring, with the crossover process simulating genetic recombination to enhance solution diversity. The mutation operation introduces random alterations to the genes of individuals, increasing variability and preventing premature convergence. Through multiple generations of evolution, the individuals in the population gradually approach the optimal solution.

The key strength of GA lies in its robust global search capability, making it highly effective for solving complex optimization problems, particularly when the solution space is large and lacks explicit patterns. GA has found widespread application in fields such as engineering optimization, machine learning, image processing, and combinatorial optimization. Although GA requires substantial computational effort and depends on appropriate parameter settings, its strong global search ability allows it to perform excellently in a wide range of practical problems.

### 2.2. Whale Optimization Algorithm (WOA)

The Whale Optimization Algorithm (WOA) is a nature-inspired optimization method based on the predatory behavior of humpback whales. This algorithm was proposed by Mirjalili and others in 2016 and is inspired by the hunting behavior of humpback whales. WOA is a global optimization algorithm [17–19], well-suited for solving a variety of complex optimization problems, and has been widely applied, particularly in the fields of multidimensional function optimization and machine learning. WOA simulates the hunting behavior of humpback whales, which includes three main strategies: encircling prey, bubble-net attacking method, and prey search.

1. Encircling Prey: Humpback whales circle around their prey and gradually reduce the size of the encircling loop. This behavior can be mathematically described as follows:

$$D = \left| C \cdot X^*(t) - X(t) \right| \tag{1}$$

$$X(t+1) = X^*(t) - A \cdot D \tag{2}$$

In this context, $X^*$ represents the current optimal solution (the position of the prey), $X$ also represents the position of the whale, $A$ and $C$ are coefficients that control the movement direction of the whale, and $D$ indicates the distance between the whale and the prey.

The definitions of coefficients $A$ and $C$ are as follows:

$$A = 2 \cdot A \cdot r - a \tag{3}$$

$$C = 2 \cdot r \tag{4}$$

Among them, *a* is a value that decreases linearly from 2 to 0, and *r* is a random number within the range of [0, 1].

2. Bubble-net Attacking Method: During the hunting process, humpback whales adopt a spiral path to approach their prey. This process can be described in the following two ways:

(1) Spiral Position Update:

$$X(t+1) = D\prime \cdot e^{bl} \cdot \cos(2\pi l) + X^*(t) \tag{5}$$

Among them, $D\prime$ is the distance between the whale and the prey, *b* is a constant, and *l* is a random number.

(2) Randomly Selecting Hunting Mode:

The next step is determined by randomly selecting between circling the prey and moving in a spiral path, deciding whether to converge toward the optimal solution or approach the prey along the spiral trajectory.

3. Search For Prey

When $|A| > 1$, the whale will move away from the current optimal solution and randomly search for other potential prey positions. This mechanism enhances the algorithm's global search ability, allowing it to escape from local optima. The mathematical model for this phase is as follows:

$$X(t+1) = X_{rand}(t) - A \cdot |C \cdot X_{rand}(t) - X(t)| \tag{6}$$

Among them, $X_{rand}$ represents the position of a randomly selected whale.

## 2.3. Construction of LSTM

Long Short-Term Memory (LSTM) is an improvement of Recurrent Neural Networks (RNN), designed to address the issues of vanishing and exploding gradients when processing long sequences [20–24]. LSTM introduces a gating mechanism, allowing it to retain important information over long time spans, making it perform exceptionally well in fields such as natural language processing and time series forecasting. The core of LSTM consists of the "cell state" and three gating mechanisms [25,26]: the forget gate, input gate, and output gate, which together control the selection, transmission, and forgetting of information,as shown in Fig 1.

1. Cell State: The cell state is the main channel in the LSTM structure, used to transmit the information flow and can carry information over long distances without significant changes. LSTM uses gating mechanisms to selectively add or delete information, allowing important information to persist over time.

2. Forget Gate: Controls whether the input at the current moment is "forgotten", that is, selecting which information to discard from the cell state. The output of the forget gate is controlled by a Sigmoid activation function with a range of 0–1.

$$f_t = \sigma(W_f \cdot [h_{t-1}, x_t] + b_f) \tag{7}$$

3. Input Gate: The input gate controls whether the current input will be added to the cell state. The input gate consists of two parts: one part decides which information will be updated, and the other part generates candidate information to update the cell state.

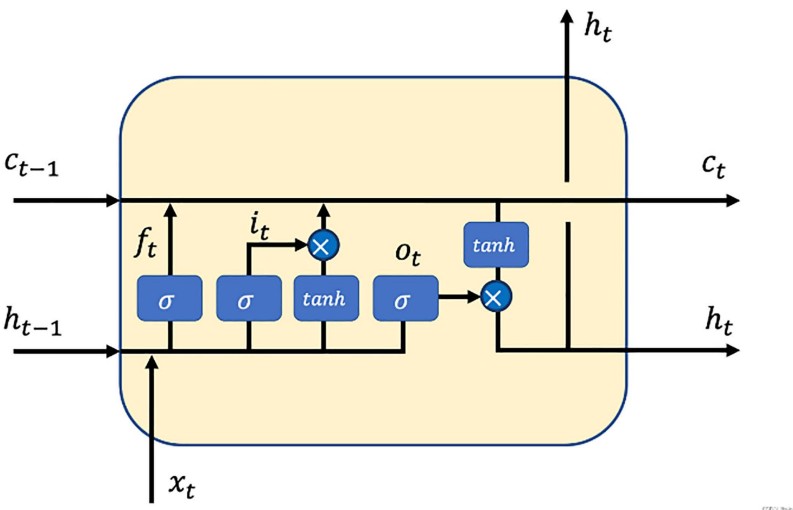

**Fig 1. LSTM structural diagram.**

(1) Input Gate Value($i_t$): Controls the proportion of the cell state that is updated.

$$i_t = \sigma(W_i \cdot [h_{t-1}, x_t] + b_i)$$

(8)

(2) Input Gate-related State Value($\tilde{C}_t$): Generates candidate information based on the current input.

$$\tilde{C}_t = \tanh(W_C \cdot [h_{t-1}, x_t] + b_C)$$

(9)

4. Cell State Update: The cell state($C_t$) update involves two parts: the forget gate deciding how much old information should be forgotten, and the input gate controlling how much new information should be added.

$$C_t = f_t \cdot C_{t-1} + i_t \cdot \tilde{C}_t$$

(10)

5. Output Gate: The output gate controls the hidden state($h_t$) at the current time step, which is used to generate the output of the LSTM. The output gate uses a Sigmoid activation function to determine the final output.

$$o_t = \sigma(W_o \cdot [h_{t-1}, x_t] + b_o)$$

(11)

The hidden state($h_t$) is obtained through a nonlinear combination of the output gate and the cell state:

$$h_t = o_t \cdot \tanh(C_t)$$

(12)

In formulas (7)–(12), Sigmoid is the activation function, and tanh is the hyperbolic tangent function. The symbols $W_f$, $W_i$, $W_c$, $W_o$ represent respectively the parameter matrices for the forget gate, input gate, input gate-related state unit, and output gate, respectively. $b_f$, $b_i$, $b_c$, $b_o$ represent the error vectors for these gates.

### 2.4. Construction of the GA-WOA-LSTM combined prediction model

**2.4.1. Background and objectives.** The stock market is characterized by non-linearity and high randomness, making traditional forecasting methods ineffective at capturing the complex market behavior [27–31]. To improve prediction accuracy, this model combines the advantages of Genetic Algorithm (GA), Whale Optimization Algorithm (WOA), and Long Short-Term Memory (LSTM) to form a hybrid optimization model (GA-WOA-LSTM). The model is designed to optimize the key parameters in time series forecasting, enabling better stock price prediction.

**2.4.2. Advantages of the combined model.** Significant performance improvement: In the GA-WOA-LSTM model proposed in this study, LSTM (long short-term memory network) can effectively capture long-term and short-term dependencies in time series data with its gating mechanism, and has advantages over traditional machine learning and deep learning models (such as BP, CNN, traditional RNN, etc.) in modeling temporal features.

Optimization strategy fusion is effective: GA (genetic algorithm) and WOA (whale optimization algorithm) are two typical intelligent optimization methods with strong global search and local fine search capabilities. Their integrated application not only improves the efficiency of hyperparameter optimization, but also effectively avoids the problem of falling into local optimum in traditional optimization process.

Robustness enhancement: GA and WOA are integrated into the training process of LSTM model, so as to realize the synergistic fusion of optimization algorithm and deep learning model, thus enhancing the tolerance of the model to noise and outliers in the data, and improving the stability and generalization ability of the prediction results.

Breaking the dilemma of local optimization: Traditional optimization methods often face the challenge of local optimization in high-dimensional complex problems. The combination of global search mechanism of GA and local dynamic update strategy of WOA makes the model have a stronger exploration ability in the search space, and effectively improves the comprehensiveness of the solution space and the global optimality of the solution Superior Comprehensive Performance: Compared to basic models such as BP, CNN, and traditional RNN, LSTM performs better in long-term dependency modeling; and compared to single-optimization-strategy models like WOA-LSTM or GA-LSTM, the GA-WOA-LSTM model, by leveraging the synergy of global and local optimization strategies, further enhances prediction accuracy, stability, and robustness, making it particularly suitable for high-noise, high-complexity prediction scenarios such as financial time series

**2.4.3. Model construction.** To improve the accuracy and stability of time series prediction, this paper proposes a hybrid forecasting model that integrates Genetic Algorithm (GA), Whale Optimizer (WOA), and Long Short-Term Memory (LSTM), as shown in Fig 2. This model achieves an effective balance between global search and local fine-tuning through a joint optimization strategy, thereby enhancing its ability to model the complex and nonlinear characteristics of financial markets.

During the model construction process, first define the hyperparameter space of the LSTM network, including the number of hidden units, Dropout ratio, batch size, and learning rate, as listed in Table 1. Each set of hyperparameters is encoded into an individual in real vector form, forming the initial population of GA. In each generation of evolution, the genetic algorithm selects fitter individuals through a tournament selection mechanism, then generates new individuals via simulated binary crossover (SBX), and maintains population diversity by incorporating Gaussian mutation strategies. Specific parameter settings are summarized in Table 2.

After several generations of genetic algorithm evolution, the model has obtained a set of parameter solutions with strong global adaptability. To further enhance the local precision of the global optimal solution obtained by GA, this study introduces the Whale Optimization Algorithm to perform refined tuning on the current elite individuals. WOA simulates the cooperative foraging behavior of humpback whales by integrating two typical strategies: "encircling prey" and "spiral updating." This enables local perturbation and deep exploitation of the optimal regions within the search space. Specifically, WOA takes the best individuals selected by GA as the center and gradually

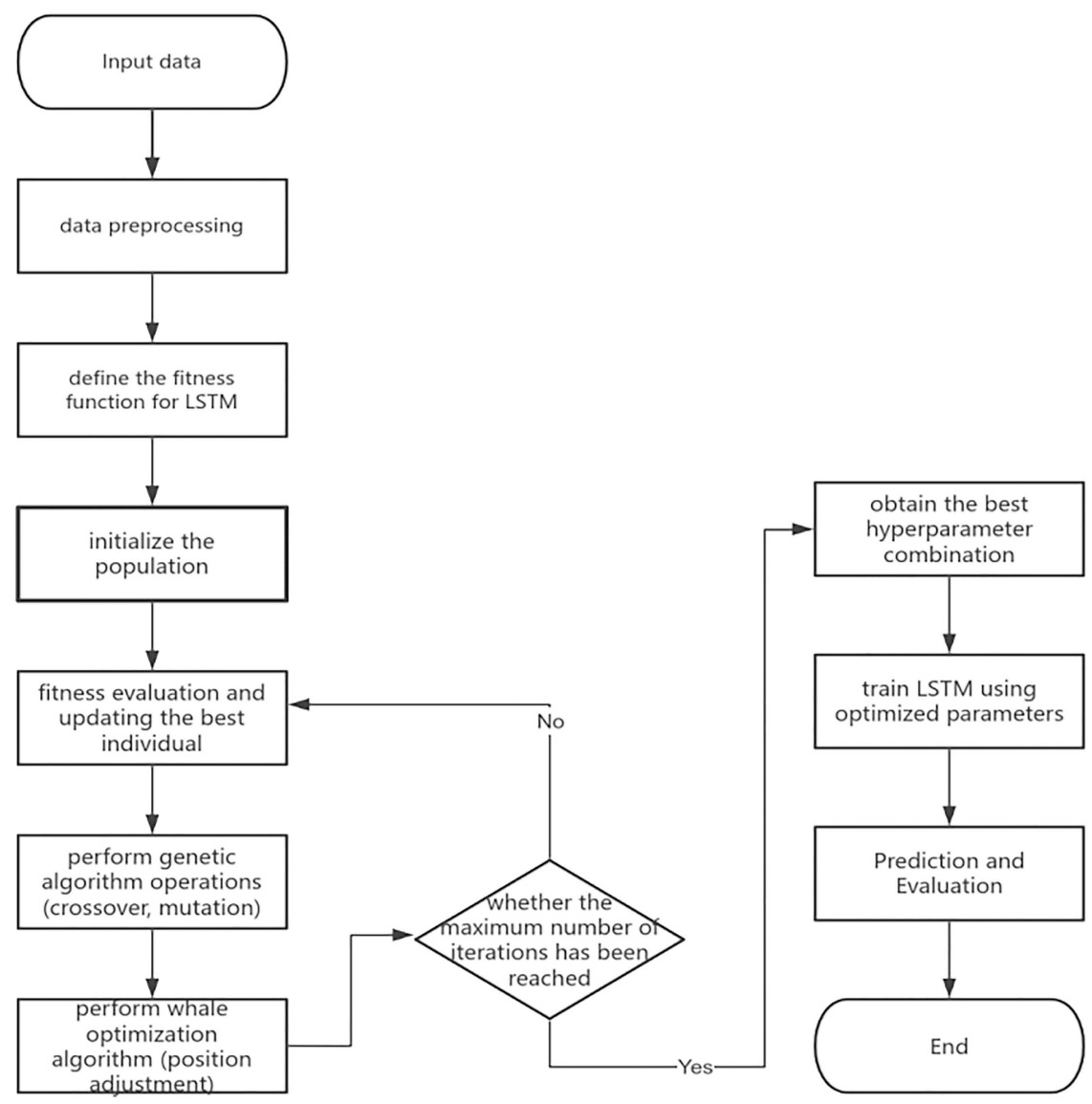

**Fig 2. Construction flow of GA-WOA-LSTM.**

**Table 1. LSTM hyperparameter configuration.**

| Hyperparameters of LSTM | Search scope |
| --- | --- |
| Units | [256, 512] |
| Dropout | [0.05, 0.15] |
| Batch size | [16, 128] |
| Learning rate | $[1 \times 10^{-4}, 1 \times 10^{-2}]$ |

**Table 2. GA parameter configuration.**

| Parameters of GA | Value/ Range |
|---|---|
| Population size | 20 |
| Number of generations | 15 |
| Mutation rate | [0.1, 0.5] |
| Fitness function | Weighted MSE + $(1 - R^2)$ |

narrows the search scope through a dynamic convergence factor, thereby capturing subtle structural variations near the local optimum and improving the model's sensitivity and adaptability to hyperparameters, as shown in Table 3. This approach not only avoids the common "premature convergence" problem of GA, but also further enhances the LSTM model's local fitting capability and generalization performance on high-noise financial time series.

The optimization process employs a hybrid cooperative framework integrating the Genetic Algorithm (GA) and the Whale Optimization Algorithm (WOA). During each generation of evolution, the population is dynamically partitioned based on a predefined fusion ratio: a subset of individuals is assigned to GA for global exploration, while the remaining individuals are refined using WOA for local exploitation. As the number of iterations increases, the allocation ratio gradually shifts towards WOA, thereby establishing a progressive transition from coarse-grained global search to fine-grained local optimization. Furthermore, an elitism strategy is incorporated to preserve the best-performing individuals in each generation and subject them to further local refinement. This mechanism ensures the stability of the optimization process and maintains diversity within the solution space, effectively mitigating the risks of performance degradation and the loss of high-quality solutions.

Finally, the optimal hyperparameters obtained through GA-WOA were used to reconstruct the LSTM prediction model, and the EarlyStopping mechanism was employed for full training. After training, the model was evaluated on both the training set and the test set, using metrics such as Mean Absolute Error (MAE), Mean Absolute Percentage Error (MAPE), Root Mean Square Error (RMSE), and coefficient of determination ($R^2$) to measure its predictive performance. The experimental results show that this hybrid optimization structure significantly enhances the model's fitting ability and stability, demonstrating strong practical application value.

## 3. Empirical analysis

### 3.1. Data sources and sample selection

This study obtained historical trading data for the trading days from March 13,2010, to March 13,2025, for three stocks: V V (stock code 600300),LOTUS HOLDINGS (stock code 600186), and SANYUAN (stock code 600429). These stocks are referred to as V V, LH, and SY respectively. The selected trading data indicators include opening price, highest price, lowest price, closing price, and trading volume. Meanwhile, the data was divided into two parts. The first 80% of samples were selected as the training set and the latter 20% as the test set. The time series samples were obtained by using the sliding window method. Each input sample was a continuous feature sequence over 8 days, with the target being the closing price on day T + 1. The original trend is shown as Figs 3–5.

**Table 3. WOA parameter configuration.**

| Parameters of WOA | Range |
|---|---|
| Convergence factor | [0, 2] |
| Attack strategy probability | [0, 1] |

## 3.2. Evaluation metrics

In this study, mean absolute error (MAE, Mean Absolute Error), root mean square error (RMSE, Root Mean Squared Error), mean absolute percentage error (MAPE, Mean Absolute Percentage Error) and coefficient of determination ($R^2$, Coefficient of Determination) were used as evaluation indicators. The formulas are shown as follows:

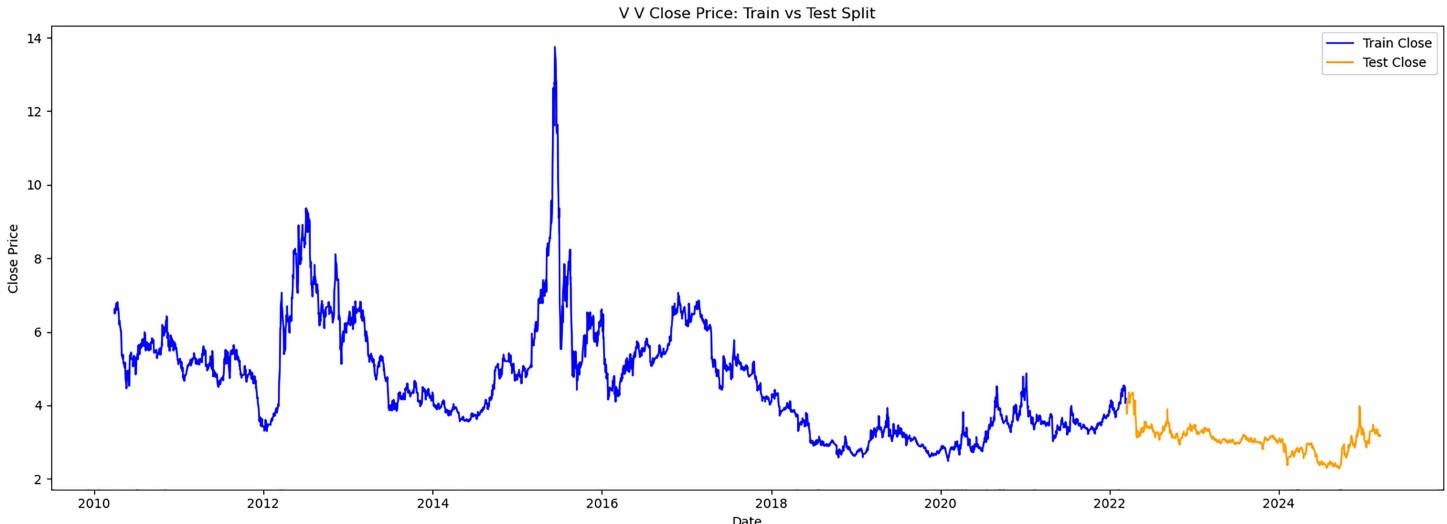

**Fig 3. V V close price.**

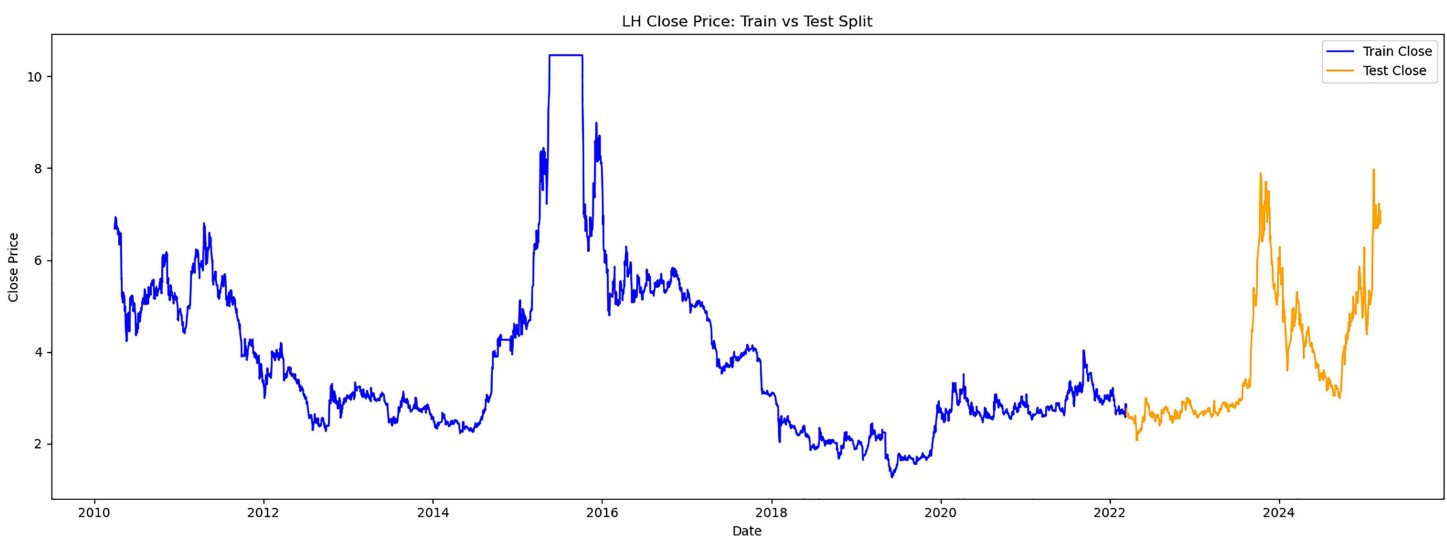

**Fig 4. LH close price.**

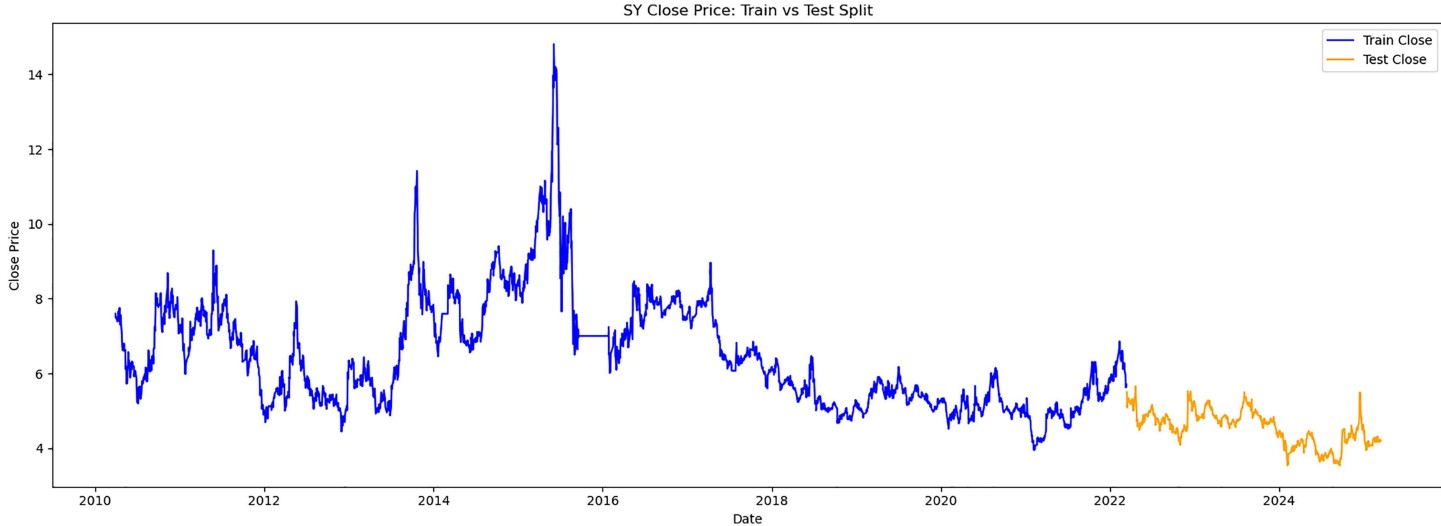

**Fig 5. SY close price.**

$$MAE = \frac{1}{n} \sum_{i=1}^{n} \left| y_i - \hat{y}_i \right| \tag{13}$$

$$RMSE = \sqrt{\frac{1}{n} \sum_{i=1}^{n} \left( y_i - \hat{y}_i \right)^2} \tag{14}$$

$$MAPE = \frac{1}{n} \sum_{i=1}^{n} \left| \frac{y_i - \hat{y}_i}{y_i} \right| \times 100\% \tag{15}$$

$$R^2 = 1 - \frac{\sum\limits_{i=1}^{n} \left( y_i - \hat{y}_i \right)^2}{\sum\limits_{i=1}^{n} \left( y_i - \overline{y} \right)^2} \tag{16}$$

MAE, RMSE and MAPE are usually used to measure the prediction ability of a model. The smaller the value is, the smaller the error is and the better the prediction effect of the model is. $R^2$ is closer to 1, indicating that the fitting degree of the model is better.

### 3.3. Closing price prediction results

In order to verify the excellence of GA-WOA-LSTM model, parameters epoch was set as 80, patience was set as 15, and validation_split was set as 0.1 to make BP, CNN, RNN, LSTM, WOA-LSTM, GA-LSTM and

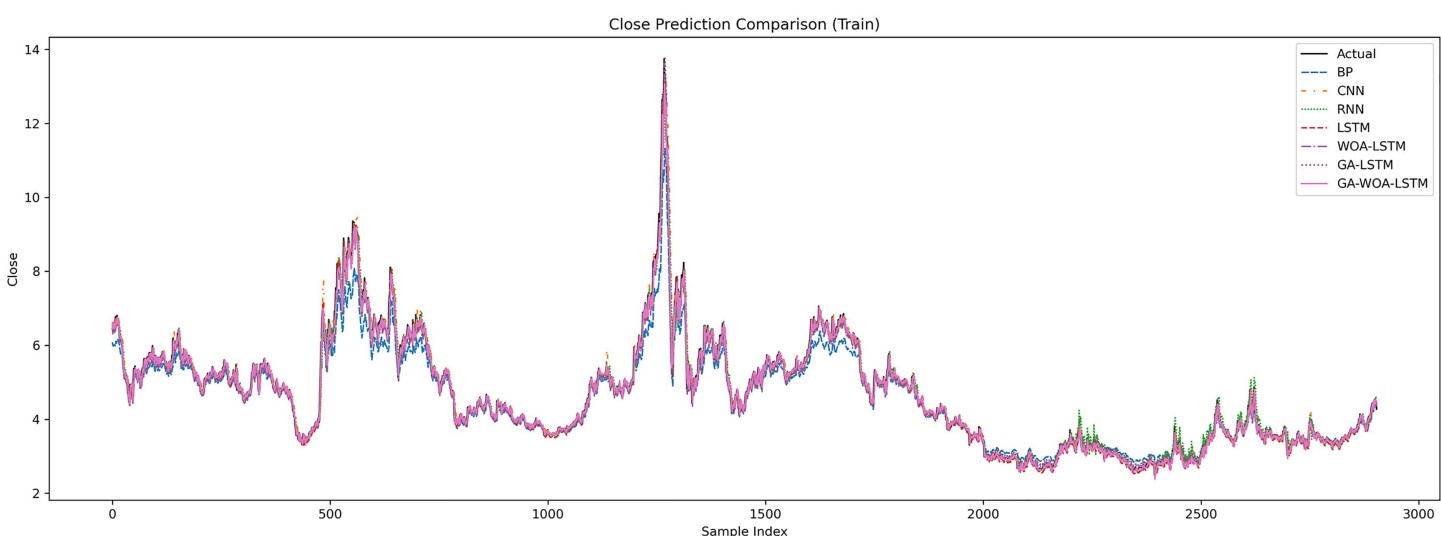

**Fig 6. Fitting results of different models on the test set of the V V dataset.**

**Fig 7. Comparative fitting results of different models on the V V train set.**

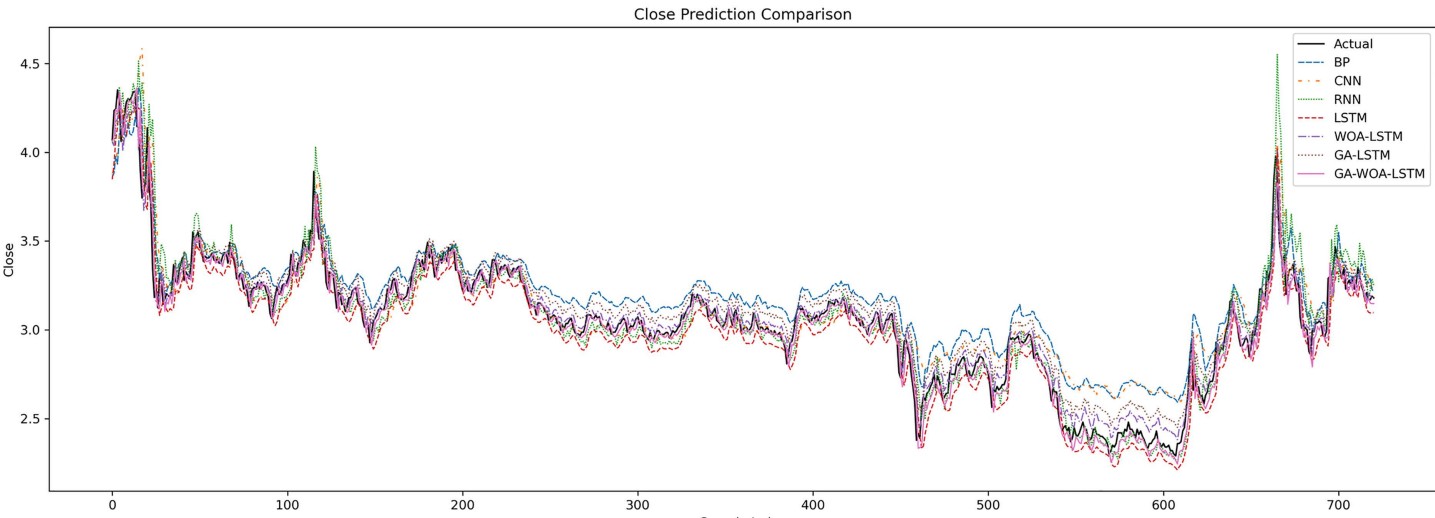

**Fig 8. Comparative fitting results of different models on the V V test set.**

## Actual vs Predicted Trends (SY)

**Fig 9. Fitting results of different models on the test set of the LH dataset.**

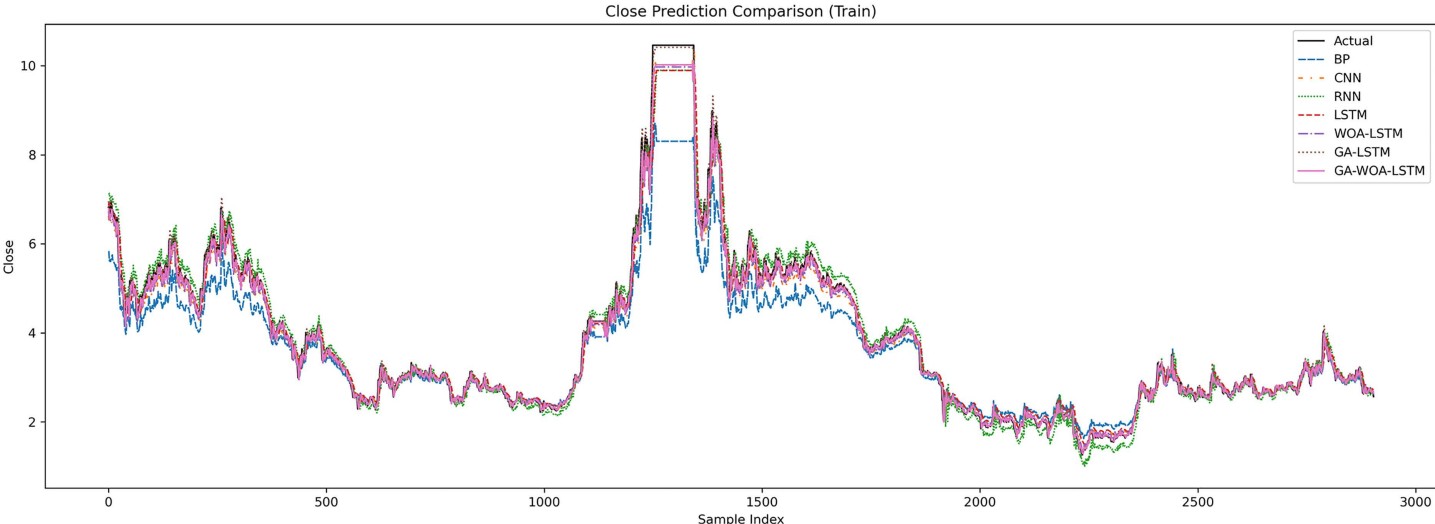

**Fig 10. Comparative fitting results of different models on the LH train set.**

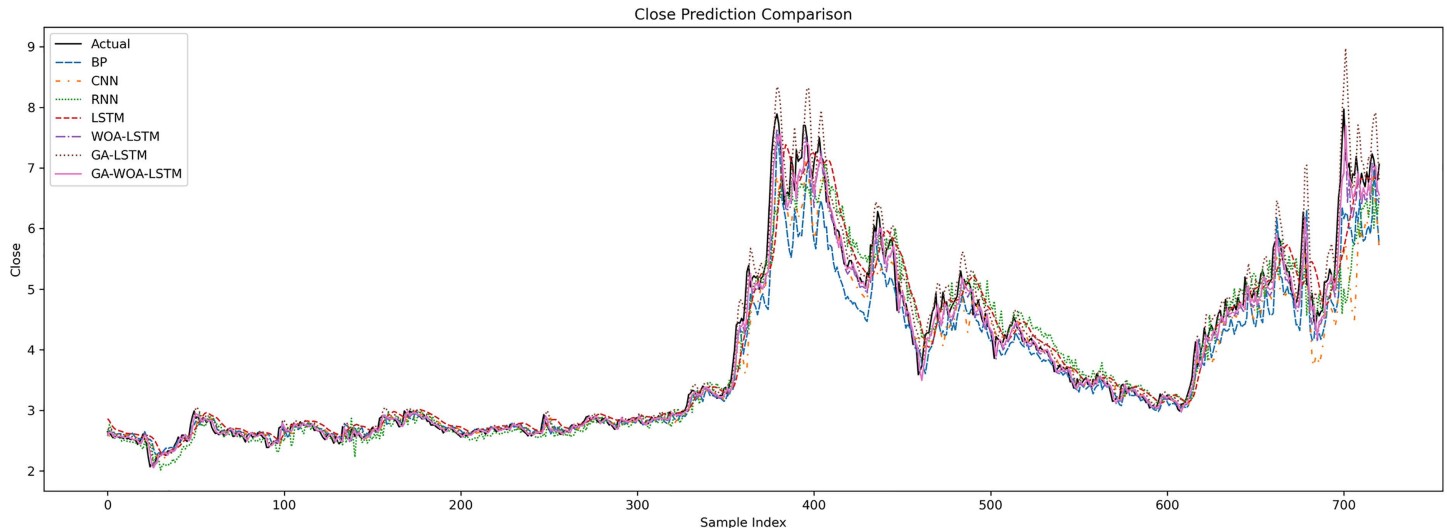

**Fig 11. Comparative fitting results of different models on the LH test set.**

GA-WOA-LSTM conduct fair experiments under the same parameters, The fitting graphs of each model are shown in Figs 6–14.

At the same time, MAE, RMSE, MAPE and R2 prediction evaluation formulas were used to calculate and evaluate the prediction results of the model, and the corresponding data values and comprehensive evaluation score curve and training loss diagram of GA-WOA-LSTM were obtained, as shown in Table 4 and Figs 15 and 16:

Actual vs Predicted Trends (SY)

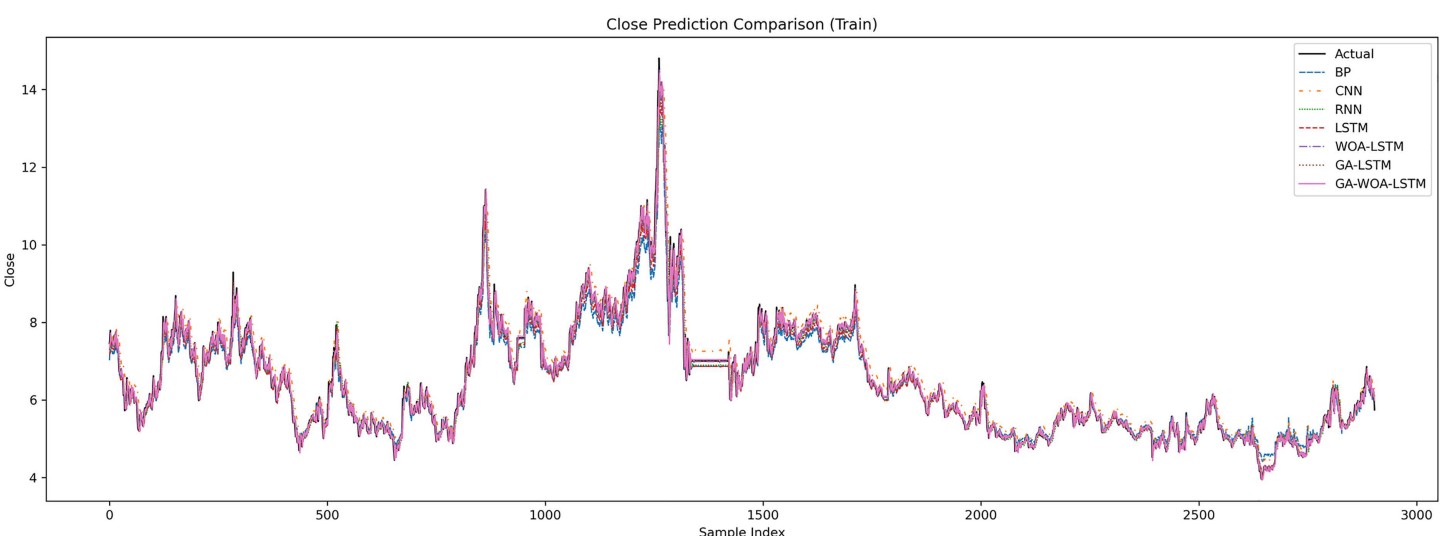

**Fig 12. Fitting results of different models on the test set of the SY dataset.**

**Fig 13. Comparative fitting results of different models on the SY train set.**

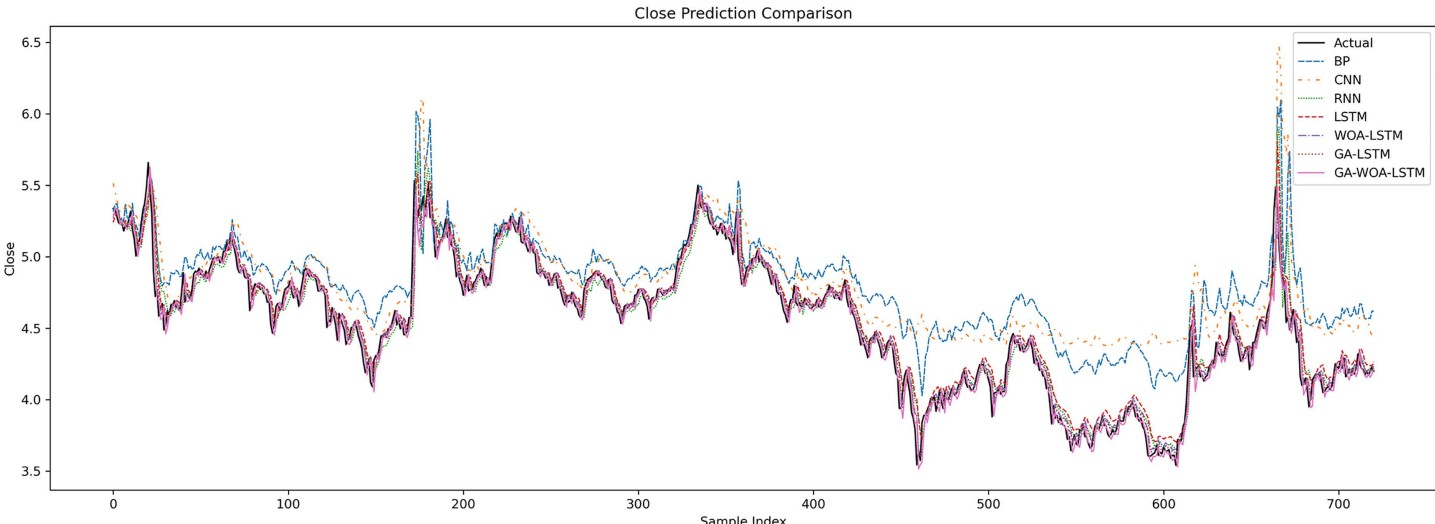

**Fig 14. Comparative fitting results of different models on the SY test set.**

**Table 4. Evaluation results of each model during training and testing.**

| Stock | Model | Training Set | | | | Test Set | | | |
|-------|-------|------|----------|------|------|------|----------|------|------|
| | | MAE | MAPE (%) | RMSE | R² | MAE | MAPE (%) | RMSE | R² |
| V V | BP | 0.2334 | 4.4092 | 0.3639 | 0.9397 | 0.1405 | 4.8648 | 0.1670 | 0.7849 |
| | CNN | 0.1467 | 3.0164 | 0.2502 | 0.9715 | 0.1105 | 3.8275 | 0.1560 | 0.8124 |
| | RNN | 0.1380 | 2.8606 | 0.2311 | 0.9757 | 0.0767 | 2.4615 | 0.1219 | 0.8854 |
| | LSTM | 0.1289 | 2.7148 | 0.1871 | 0.9840 | 0.0918 | 3.0092 | 0.1120 | 0.9033 |
| | WOA-LSTM | 0.1188 | 2.3500 | 0.1769 | 0.9857 | 0.0535 | 1.7901 | 0.0734 | 0.9584 |
| | GA-LSTM | 0.1109 | 2.3358 | 0.1634 | 0.9878 | 0.0907 | 3.0937 | 0.1067 | 0.9121 |
| | **GA-WOA-LSTM** | 0.0968 | 1.9466 | 0.1616 | 0.9881 | **0.0476** | **1.5523** | **0.0706** | **0.9616** |
| LH | BP | 0.3599 | 7.5173 | 0.5852 | 0.9049 | 0.2372 | 4.9397 | 0.3920 | 0.9208 |
| | CNN | 0.1530 | 3.7038 | 0.2281 | 0.9856 | 0.2176 | 4.6144 | 0.4016 | 0.9168 |
| | RNN | 0.2091 | 5.5732 | 0.2899 | 0.9767 | 0.2284 | 5.1770 | 0.3915 | 0.9210 |
| | LSTM | 0.1672 | 4.3280 | 0.2704 | 0.9797 | 0.2099 | 4.9177 | 0.3462 | 0.9382 |
| | WOA-LSTM | 0.1068 | 2.5048 | 0.1675 | 0.9922 | 0.1177 | 2.6304 | 0.1956 | 0.9803 |
| | GA-LSTM | 0.0832 | 2.2453 | 0.1307 | 0.9953 | 0.1570 | 3.6172 | 0.2586 | 0.9655 |
| | **GA-WOA-LSTM** | 0.0907 | 2.1004 | 0.1536 | 0.9934 | **0.1088** | **2.4744** | **0.1842** | **0.9825** |
| SY | BP | 0.2046 | 2.9160 | 0.2937 | 0.9603 | 0.2755 | 6.4113 | 0.3258 | 0.4894 |
| | CNN | 0.2552 | 3.9043 | 0.3579 | 0.9411 | 0.2647 | 6.2306 | 0.3465 | 0.4223 |
| | RNN | 0.1618 | 2.3451 | 0.2485 | 0.9716 | 0.0799 | 1.7598 | 0.1195 | 0.9313 |
| | LSTM | 0.1618 | 2.3234 | 0.2353 | 0.9746 | 0.0799 | 1.7971 | 0.1125 | 0.9391 |
| | WOA-LSTM | 0.1146 | 1.6904 | 0.1790 | 0.9853 | 0.0592 | 1.3125 | 0.0855 | 0.9648 |
| | GA-LSTM | 0.1159 | 1.7168 | 0.1827 | 0.9847 | 0.0612 | 1.3506 | 0.0885 | 0.9623 |
| | **GA-WOA-LSTM** | 0.1140 | 1.6936 | 0.1778 | 0.9855 | **0.0581** | **1.2802** | **0.0847** | **0.9655** |

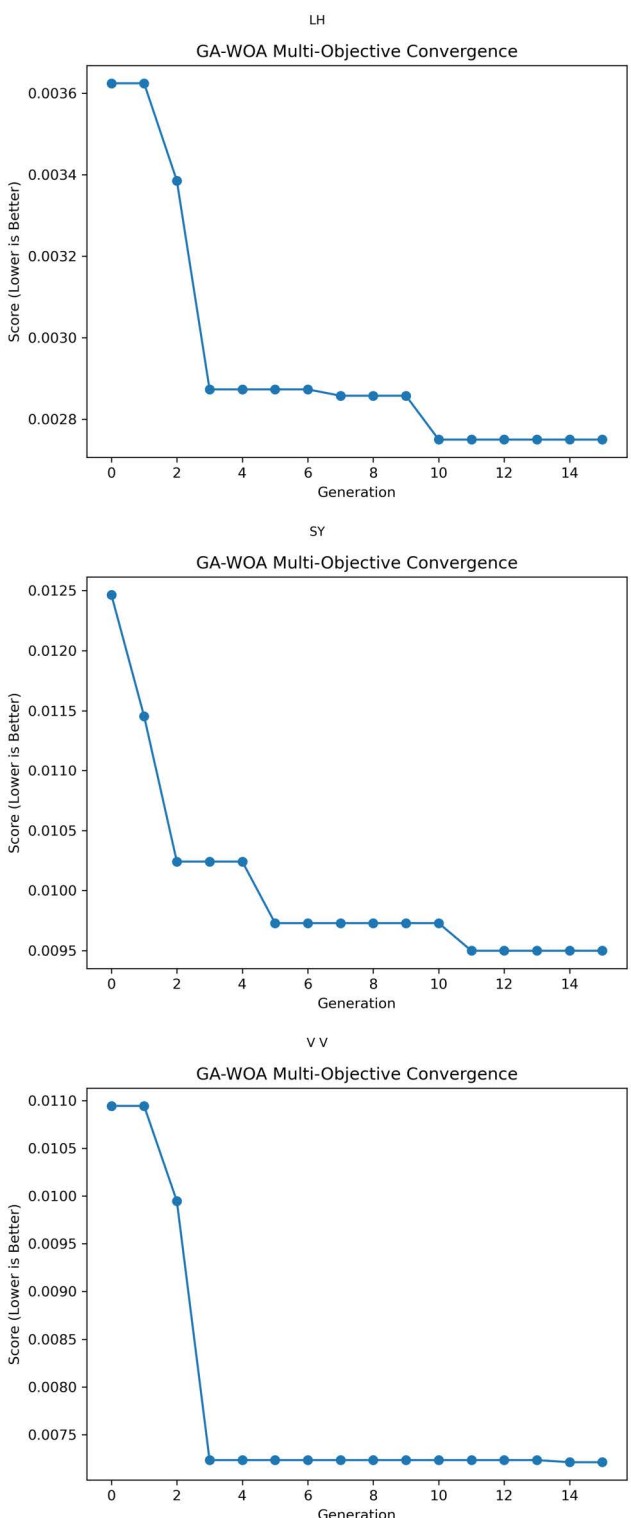

**Fig 15. Comprehensive evaluation score curve.**

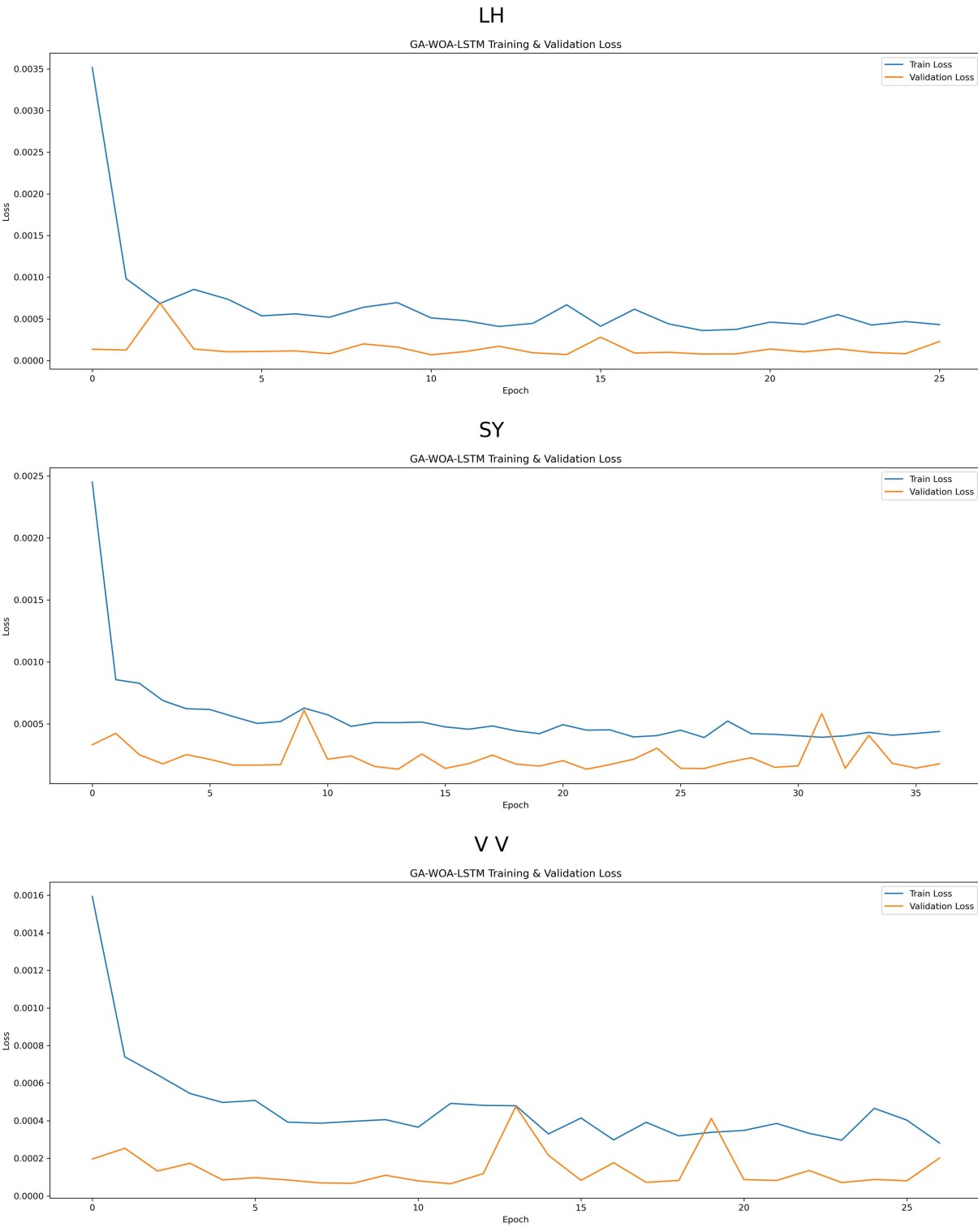

**Fig 16. Training loss diagram of GA-WOA-LSTM.**

V V Model Evaluation Metrics Comparison

**Fig 17. V V Performance comparison of different prediction models across evaluation metrics.**

From Table 4 and Figs 17–19, it can be seen that the prediction results of each stock using GA-WOA-LSTM are superior to those of the other six models. This indicates that the GA-WOA-LSTM ensemble model performed the best among these seven models in this test. Additionally, the low error and high fit of GA-WOA-LSTM suggest its practicality. In the field of stock prediction, especially in capturing short-term closing price trends, this model can provide investors with more reliable decision support.

## 4. Conclusion

This study proposes a hybrid forecasting model that integrates Genetic Algorithm (GA), Whale Optimization Algorithm (WOA), and Long Short-Term Memory neural network (LSTM), referred to as GA-WOA-LSTM, to address the nonlinear, high-noise, and dynamic characteristics commonly present in stock market time series data. The model employs GA to conduct a global search of LSTM hyperparameters and incorporates WOA for local fine-tuning, thereby effectively enhancing the prediction accuracy and stability of the model.

In the empirical analysis, the proposed model was validated on multiple stock closing price datasets. Experimental results demonstrate that GA-WOA-LSTM outperforms traditional models such as BP, CNN, RNN, and LSTM, as well as models based on single optimization algorithms, in terms of key evaluation metrics including MAE, MAPE, RMSE, and $R^2$. Notably, the model shows superior fitting performance and generalization capability on the test sets. These findings

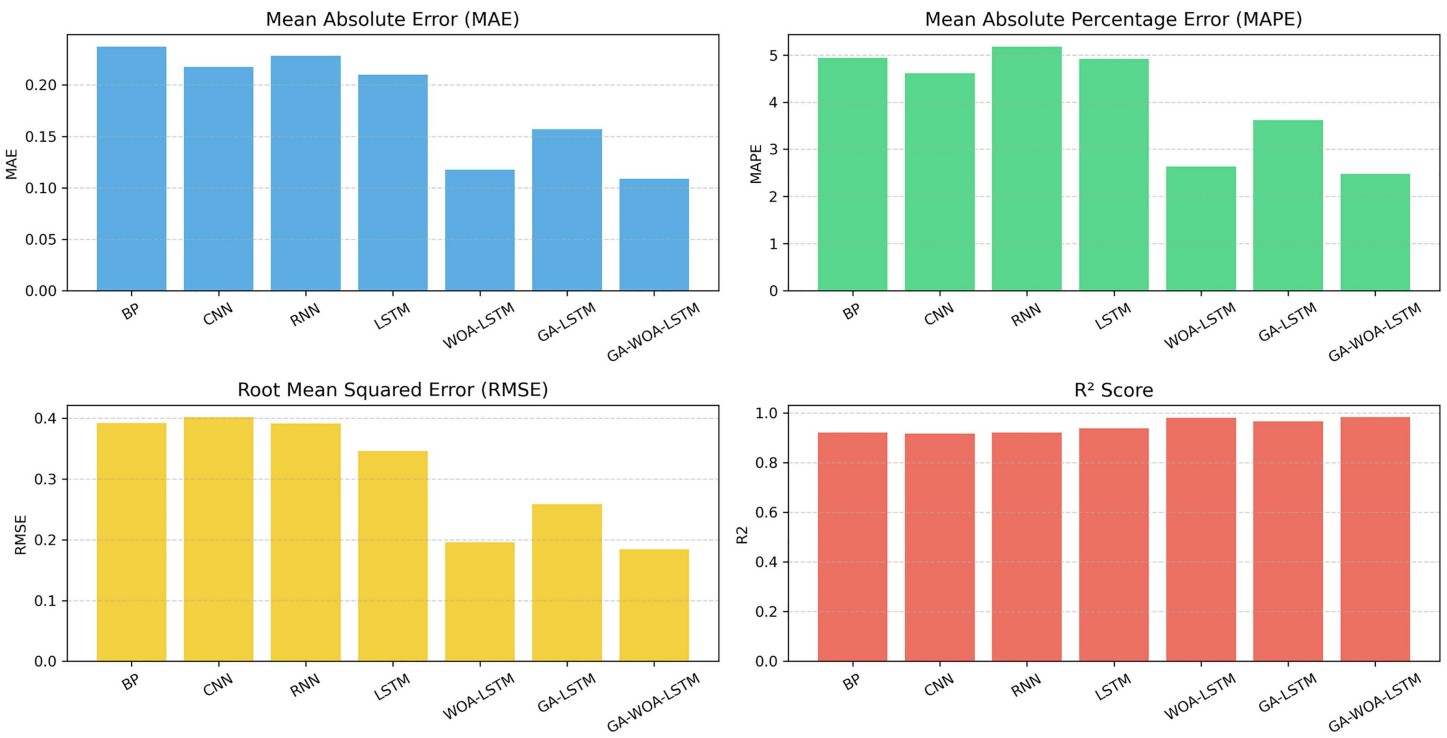

**Fig 18. LH Performance comparison of different prediction models across evaluation metrics.**

indicate that the hybrid optimization strategy can effectively avoid local optima during hyperparameter search and significantly improve the model's responsiveness to market volatility.

Although the GA-WOA-LSTM model demonstrated good prediction performance in this study, there are still some shortcomings. Firstly, stock market prediction does not solely rely on historical price data but is also influenced by various external factors such as macroeconomic conditions, market sentiment, and more. Future research could attempt to incorporate additional external variables into the model, such as news sentiment analysis or policy changes, to further improve prediction accuracy. Secondly, while LSTM performs well in handling time series data, it may face issues like vanishing or exploding gradients when dealing with longer time series data. Future studies could consider incorporating other deep learning models, such as GRU (Gated Recurrent Unit) or Transformer, to improve the model's long-term memory capabilities.

Overall, the GA-WOA-LSTM model proposed in this paper demonstrates high effectiveness in stock market prediction and offers new insights for future research in this area. With further optimization of the model and integration of external data, it is expected to achieve more accurate and robust stock market predictions, providing more reliable decision support for investors and financial institutions.

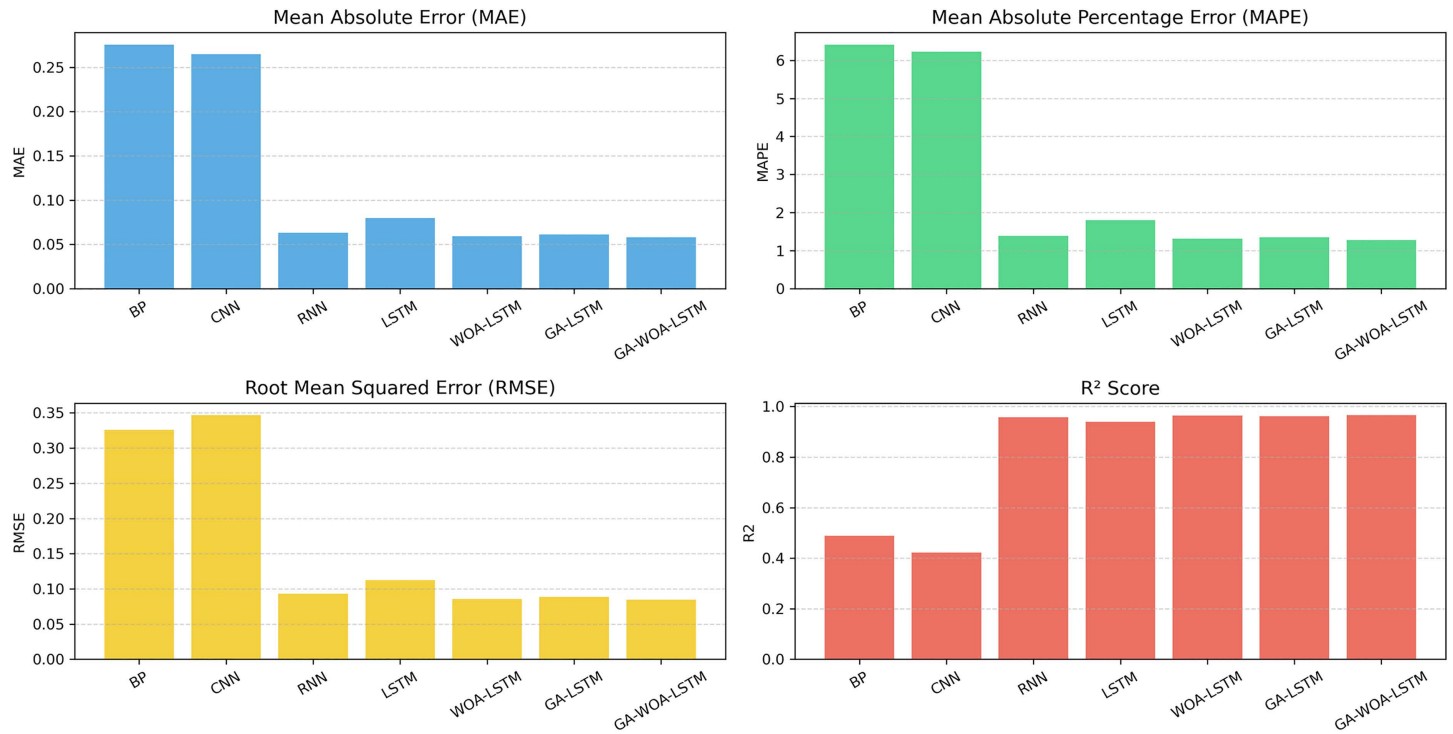

**Fig 19. SY Performance comparison of different prediction models across evaluation metrics.**

## Supporting information

**S1 File. Paper model.**

(ZIP)

## Author contributions

**Data curation:** Zunlong Wang.

**Formal analysis:** Zunlong Wang.

**Methodology:** Wu Huiyong.

**Supervision:** Wu Huiyong.

**Writing – original draft:** Zunlong Wang.

**Writing – review & editing:** Zunlong Wang.

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
