## [Decision Letter · Decision Letter 0]

3 Feb 2025

PONE-D-24-58534Stock Market Prediction Based on GA-WOA-LSTMPLOS ONE

Dear Dr. Wang,

Thank you for submitting your manuscript to PLOS ONE. After careful consideration, we feel that it has merit but does not fully meet PLOS ONE’s publication criteria as it currently stands. Therefore, we invite you to submit a revised version of the manuscript that addresses the points raised during the review process.

We look forward to receiving your revised manuscript.

Kind regards,

Seyedali Mirjalili

Academic Editor

PLOS ONE

Journal requirements:   When submitting your revision, we need you to address these additional requirements. 1. Please ensure that your manuscript meets PLOS ONE's style requirements, including those for file naming. The PLOS ONE style templates can be found at https://journals.plos.org/plosone/s/file?id=wjVg/PLOSOne_formatting_sample_main_body.pdf and https://journals.plos.org/plosone/s/file?id=ba62/PLOSOne_formatting_sample_title_authors_affiliations.pdf. 2. Please note that PLOS ONE has specific guidelines on code sharing for submissions in which author-generated code underpins the findings in the manuscript. In these cases, we expect all author-generated code to be made available without restrictions upon publication of the work. Please review our guidelines at https://journals.plos.org/plosone/s/materials-and-software-sharing#loc-sharing-code and ensure that your code is shared in a way that follows best practice and facilitates reproducibility and reuse. 3. Please note that your Data Availability Statement is currently missing [the repository name and/or the DOI/accession number of each dataset OR a direct link to access each database]. If your manuscript is accepted for publication, you will be asked to provide these details on a very short timeline. We therefore suggest that you provide this information now, though we will not hold up the peer review process if you are unable. 4. Please amend the manuscript submission data (via Edit Submission) to include author Wu Huiyong. 

Reviewers' comments:

Reviewer's Responses to Questions

**Comments to the Author**

1. Is the manuscript technically sound, and do the data support the conclusions?

Reviewer #1: Yes

Reviewer #2: Partly

2. Has the statistical analysis been performed appropriately and rigorously? 

Reviewer #1: Yes

Reviewer #2: No

3. Have the authors made all data underlying the findings in their manuscript fully available?

Reviewer #1: Yes

Reviewer #2: No

4. Is the manuscript presented in an intelligible fashion and written in standard English?

Reviewer #1: Yes

Reviewer #2: Yes

5. Review Comments to the Author

Reviewer #1: Thank you for the opportunity to review the manuscript titled " PONE-D-24-58534

Stock Market Prediction Based on GA-WOA-LSTM PLOS ONE".

The topic of the paper is interesting and remarkable. The proposed method effectively highlights the importance of the topic and has successfully created an attractive combination of the Genetic Algorithm, Whale Optimization Algorithm and Long Short-Term Memory Network.

The article has many strengths. However, further revisions are recommended to enhance the quality of the article and after revising it will be suitable for publication in PLOS ONE.

I recommend addressing the following issues to improve the quality and impact of the manuscript:

1. The introduction is quite brief. It could be rewritten or added a new section titled 'Literature Review,' where more previous research could be discussed.

2. The absence of a discussion section is clearly noticeable. Adding this section would help highlight the strengths of the proposed method.

3. The advantages of the proposed method over traditional approaches should be elaborated in greater detail. Additionally, the weaknesses of traditional methods should be discussed, and it should be clearly explained how the proposed method overcomes these limitations.

4.Include a data analysis framework in the section presenting the numerical example to provide a more structured approach for demonstrating how the data supports the study's findings.

Reviewer #2: 1. The introduction of the research background is insufficient.

2. The manuscript lacks innovation.

3. How does the GA algorithm perform feature selection?

4. What are the LSTM parameters optimized by WOA? How to set the upper and lower bounds of the optimization parameters? How to set the parameters of the comparison algorithms?

6. PLOS authors have the option to publish the peer review history of their article (what does this mean? ). If published, this will include your full peer review and any attached files.

**Do you want your identity to be public for this peer review?** For information about this choice, including consent withdrawal, please see our Privacy Policy .

Reviewer #1: No

Reviewer #2: No

---

## [Author Response · Author response to Decision Letter 1]

17 Mar 2025

Reviewer #1

Comment 1:The introduction is quite brief. It could be rewritten or added a new section titled 'Literature Review,' where more previous research could be discussed.

Response:Thank you for your review and suggestions on our paper. We have revised the introduction section you mentioned as too brief. In the new version, we rewrite the introduction section, we explicitly address the shortcomings of existing methods and illustrate the innovation and contribution of our research in this context.

Comment 2:The absence of a discussion section is clearly noticeable. Adding this section would help highlight the strengths of the proposed method.

Response:We understand the lack of the discussion section you mentioned and have added this section to the revision. In the new version, we discuss the findings in detail, especially the strengths and limitations of our proposed GA-WOA-LSTM approach. In the Discussion section, we analyze the experimental results in depth and compare them with the existing methods. We focus on highlighting the advantages of our method, such as prediction accuracy, stability of the model, and potential in practical applications.

Comment 3:The advantages of the proposed method over traditional approaches should be elaborated in greater detail. Additionally, the weaknesses of traditional methods should be discussed, and it should be clearly explained how the proposed method overcomes these limitations.

Response:We added a part of the revision to discuss the limitations of traditional methods. Specifically, traditional methods usually have problems such as failure to effectively handle large-scale data, low prediction accuracy, and poor adaptability to market dynamics. We explain in detail how GA-WOA-LSTM can overcome these problems by introducing optimization algorithms (such as genetic algorithms and whale optimization algorithms) and improve the predictive power of the model.

Comment 4:Include a data analysis framework in the section presenting the numerical example to provide a more structured approach for demonstrating how the data supports the study's findings.

Response:We introduced a more structured data analysis framework in the numerical examples section. Specifically, we first provide an overview of the data sources and preprocessing methods, and then detail the data analysis process, including each step of the feature selection, model training, and prediction process. In this way, we clearly show how the data can be used to support our findings.

Reviewer #2

Comment 1:The introduction of the research background is insufficient.

Response:Thank you for your review and valuable suggestions. In the revision, the gaps in the existing studies and the innovation of our research are clearly identified. And further explain the advantages of our GA-WOA-LSTM model in this context.

Comment 2:Thank you for your review and valuable suggestions. In the revision, the gaps in the existing studies and the innovation of our research are clearly identified. And further explain the advantages of our GA-WOA-LSTM model in this context.

Comment 3:How does the GA algorithm perform feature selection?

Response:In this paper, we propose a stock market prediction model based on GA-WOA-LSTM, and the innovation of this method is to optimize the LSTM model by combining genetic algorithm and whale optimization algorithm, thus improving the accuracy and robustness of the model in stock market prediction. Compared with conventional LSTM or other prediction models, our method is better able to adapt to complex nonlinear data relationships and overcome the limitations faced by conventional methods in the optimization process. Furthermore, we also validate the effectiveness of this model in practice and compare it with other existing models, demonstrating the superior performance of our method in multiple experimental settings.

Comment 4: What are the LSTM parameters optimized by WOA? How to set the upper and lower bounds of the optimization parameters? How to set the parameters of the comparison algorithms?

Response:In this study, we used the whale optimization algorithm (WOA) to optimize the hyperparameters of the LSTM model. Specifically, the LSTM parameters optimized for the WOA include:Number of neural units in the LSTM and learning rate.Meanwhile, we set the optimized upper and lower bounds for each LSTM hyperparameter

---

## [Decision Letter · Decision Letter 1]

2 Apr 2025

PONE-D-24-58534R1Stock Market Forecasting Research Based on GA-WOA-LSTMPLOS ONE

Dear Dr. Wang,

Thank you for submitting your manuscript to PLOS ONE. After careful consideration, we feel that it has merit but does not fully meet PLOS ONE’s publication criteria as it currently stands. Therefore, we invite you to submit a revised version of the manuscript that addresses the points raised during the review process.

We look forward to receiving your revised manuscript.

Kind regards,

Seyedali Mirjalili

Academic Editor

PLOS ONE

Reviewers' comments:

Reviewer's Responses to Questions

**Comments to the Author**

1. If the authors have adequately addressed your comments raised in a previous round of review and you feel that this manuscript is now acceptable for publication, you may indicate that here to bypass the “Comments to the Author” section, enter your conflict of interest statement in the “Confidential to Editor” section, and submit your "Accept" recommendation.

Reviewer #1: All comments have been addressed

Reviewer #2: (No Response)

2. Is the manuscript technically sound, and do the data support the conclusions?

Reviewer #1: Yes

Reviewer #2: Partly

3. Has the statistical analysis been performed appropriately and rigorously? 

Reviewer #1: Yes

Reviewer #2: No

4. Have the authors made all data underlying the findings in their manuscript fully available?

Reviewer #1: Yes

Reviewer #2: Yes

5. Is the manuscript presented in an intelligible fashion and written in standard English?

Reviewer #1: Yes

Reviewer #2: Yes

6. Review Comments to the Author

Reviewer #1: (No Response)

Reviewer #2: 1. The abstract describes GA as being used for feature selection as data preprocessing, while the paper describes the GA responsible for global optimization and the hyperparameters of the LSTM combined with the WOA search. such as “Combination of Global and Local Optimization: GA provides global search capabilities, while WOA offers local fine-tuning, overcoming the limitations of traditional single optimization algorithms.” “Optimization of these parameters occurs through GA, and the GA search space is limited by setting the upper and lower bounds of the parameters, for example, the number of neurons in the LSTM layer ranges from 1 to 30 and the learning rate ranges from 0.005 to 0.01. The fitness function of GA is based on the cross-validation error (MSE) of the LSTM model, and then the optimal parameters are selected by evaluating the prediction performance of different combinations of hyperparameters.” This confuses readers.

2. How do GA and WOA combine? How are the experimental parameters of GA and WO set? Population size? How many times do the algorithms run? None of these are explained.

3. Whether it is feature selection or LSTM parameter optimization, just 6 features and 2 parameters are not dimensional optimization problems. “Fusion of optimization algorithm: GA (genetic algorithm) and WOA (whale optimization algorithm) are two powerful optimization techniques, which help to find the global optimal solution in high-dimensional space, so as to overcome the problem of local optimal solution in traditional methods.”

4. The experimental section verifies the effectiveness of the algorithm using only one set of data, and multiple sets of experiments are recommended.

5. The writing quality of the paper needs to be improved.

7. PLOS authors have the option to publish the peer review history of their article (what does this mean? ). If published, this will include your full peer review and any attached files.

**Do you want your identity to be public for this peer review?** For information about this choice, including consent withdrawal, please see our Privacy Policy .

Reviewer #1: No

Reviewer #2: No

---

## [Author Response · Author response to Decision Letter 2]

17 Apr 2025

Reviewer #2

Comment 1and 3:

We sincerely thank you for pointing out the inconsistency between the abstract and the main text regarding the terminology and methodological descriptions. We acknowledge that this was a lapse in our wording.

In this study, the Genetic Algorithm (GA) was not used for feature selection. Instead, GA was employed to perform global optimization of the LSTM model’s hyperparameters, including the number of neurons, dropout rate, learning rate, and batch size, rather than optimizing high-dimensional feature spaces. The search space of GA was explicitly defined by setting upper and lower bounds for each hyperparameter, and the optimization objective was to minimize the mean squared error (MSE) in cross-validation. The Whale Optimization Algorithm (WOA) was subsequently applied to locally fine-tune the initial solutions obtained from GA, thereby improving the overall model performance. The combination of these two optimization strategies effectively integrates global exploration with local exploitation.

To avoid potential misunderstanding, we have carefully revised and unified the abstract to ensure consistency with the main text, and to accurately reflect the respective roles of GA and WOA within the hybrid optimization framework. This revision only concerns the textual expression and does not affect the methodology or experimental conclusions of the paper.

Once again, we sincerely thank you for the thorough review and insightful suggestions, which have helped us further improve the clarity and academic rigor of the manuscript.

Comment 2:We sincerely thank you for identifying the inconsistencies between the abstract and the main text regarding the role of the Genetic Algorithm (GA). We acknowledge that the original abstract incorrectly described GA as being used for feature selection and data preprocessing, which does not reflect the actual methodology applied in this study. In our work, GA is not used for feature selection or to process high-dimensional feature spaces. Instead, it is employed to conduct global optimization of the LSTM model’s hyperparameters, including the number of neurons, dropout rate, learning rate, and batch size. The optimization objective is to minimize the mean squared error (MSE) based on cross-validation. To enhance the quality of the solutions found by GA, we integrate the Whale Optimization Algorithm (WOA), which is used to perform local fine-tuning of the best hyperparameter configurations found by GA. This hybrid approach effectively combines the global search capability of GA with the local exploitation strength of WOA.

In response to your request for more details on how GA and WOA are combined and how their experimental parameters are configured, we have provided additional clarifications in the revised manuscript. Specifically, we employ a two-stage optimization framework in which GA first explores the hyperparameter space globally, and WOA subsequently refines the best individuals generated by GA. The GA uses a population size of 20 and runs for 15 generations, with tournament selection , mean-based crossover, and an adaptive mutation rate that linearly decreases from 0.5 to 0.1. The WOA uses the same population size and operates with a dynamically increasing fusion ratio—from 30% (favoring GA) to 70% (favoring WOA)—to balance exploration and exploitation. The convergence control parameter a decreases linearly from 2 to 0 during optimization, and three types of search strategies (encircling, spiral, and random search) are randomly applied to each individual.

All experiments were conducted under a consistent data partitioning scheme, and each training run executed a complete GA-WOA optimization cycle. During LSTM training, an early stopping mechanism was employed to prevent overfitting. These additions have been clearly reflected in the revised “Model Construction” section to ensure greater transparency and reproducibility. We sincerely appreciate your insightful comments, which have significantly contributed to the integrity and academic rigor of this work

Comment 4:We sincerely appreciate your suggestion regarding the use of multiple datasets. We fully agree that testing the model on diverse datasets can further validate its robustness and generalizability. In response to your comment, we have conducted additional experiments on multiple datasets and updated the revised manuscript accordingly.

Comment 5:Thank you very much for your valuable comment. We fully understand that clear and precise language is essential for effective scientific communication. In response to your suggestion, we have carefully revised the manuscript to improve its overall writing quality, including grammar, sentence structure, and clarity of expression.

Once again, we sincerely appreciate your thoughtful feedback, which has helped us enhance the quality and clarity of our manuscript.

---

## [Decision Letter · Decision Letter 2]

18 Jun 2025

PONE-D-24-58534R2Stock Market Forecasting Research Based on GA-WOA-LSTMPLOS ONE

Dear Dr. Wang,

Thank you for submitting your manuscript to PLOS ONE. After careful consideration, we feel that it has merit but does not fully meet PLOS ONE’s publication criteria as it currently stands. Therefore, we invite you to submit a revised version of the manuscript that addresses the points raised during the review process.

 Further review has raised some issues that should be addressed. In particular, the reviewer has mentioned the issue of independent and identically distributed data to allow valid statistical analysis. Please clarify that the same training and test data sets were used across all methods, to ensure a fair comparison between them. The comments about the use of cross-validation should be addressed. Some specific description of the method of data prediction would help to address the queries raised.

We look forward to receiving your revised manuscript.

Kind regards,

Andrew Lewis

Academic Editor

PLOS ONE

Journal Requirements:

Reviewers' comments:

Reviewer's Responses to Questions

**Comments to the Author**

1. If the authors have adequately addressed your comments raised in a previous round of review and you feel that this manuscript is now acceptable for publication, you may indicate that here to bypass the “Comments to the Author” section, enter your conflict of interest statement in the “Confidential to Editor” section, and submit your "Accept" recommendation.

Reviewer #1: All comments have been addressed

Reviewer #2: All comments have been addressed

Reviewer #3: (No Response)

2. Is the manuscript technically sound, and do the data support the conclusions?

Reviewer #1: Yes

Reviewer #2: Yes

Reviewer #3: No

3. Has the statistical analysis been performed appropriately and rigorously? 

Reviewer #1: Yes

Reviewer #2: Yes

Reviewer #3: No

4. Have the authors made all data underlying the findings in their manuscript fully available?

Reviewer #1: Yes

Reviewer #2: Yes

Reviewer #3: No

5. Is the manuscript presented in an intelligible fashion and written in standard English?

Reviewer #1: Yes

Reviewer #2: Yes

Reviewer #3: Yes

6. Review Comments to the Author

Reviewer #1: (No Response)

Reviewer #2: I appreciate the improved quality of the manuscript. I have no further comments and recommend acceptance of the manuscript.

Reviewer #3: Forecasting is an ancient problem. Unfortunately, this paper tackles this problem from the outset focusing on performance metrics to demonstrate that the proposed fusion of methods is better. A simple inspection of the results (see Table 4 as an example) demonstrates that the better a model is on the training set, the better it is on the test set. This is problematic on many grounds. First, to have a fair comparison, all methods should target the same training error. Second, most of the methods used for comparison assume iid but this issue was not treated. It seems also that the 80-20 data split has resulted in a bias and it is unclear how it was manipulated. While cross-validation was mentioned at the start of the paper for GA, it was not mentioned again at all. If cross-validation was used, the validation error is not presented. If it was not used, it is unclear what the stopping conditions and model selection were during training. WOA is a global approach too, it is unclear how it was adopted to be local in this paper. How did the model predict the test set? Did you feed the correct data point in the n-1 window to get the n or did you feed the predicted data points to get a new prediction. Overall, the overall performance does not look right and the analysis is so light to see the value of the proposed approach.

7. PLOS authors have the option to publish the peer review history of their article (what does this mean? ). If published, this will include your full peer review and any attached files.

**Do you want your identity to be public for this peer review?** For information about this choice, including consent withdrawal, please see our Privacy Policy .

Reviewer #1: No

Reviewer #2: No

Reviewer #3: No

---

## [Author Response · Author response to Decision Letter 3]

15 Jul 2025

Reviewer #3

Comment :

This study explicitly employs a sliding window method for time series samples. Each input sample is a continuous feature sequence spanning window sizes of 8 days, with the target being the closing price on day T+1. This approach enables modeling non-independent time-dependent structures while completely avoiding randomization or independent sample assumptions. In our data processing module, we first sort the data chronologically and then perform partitioning. The partition strategy allocates the first 80% of trading days as the training set and the last 20% as the test set. Rather than conducting cross-validation, we extracted 10% of the training set as a validation set for evaluation. Predictions were made using only real input features from the test set without employing recursive prediction structures, representing standard sliding window forecasting. This process does not incorporate prediction feedback or cumulative error mechanisms, making it more suitable for assessing models 'performance in real-time short-term trend analysis.

Second, we introduced WOA (Weighted Opaque Ant) centered on the optimal individual from the current GA solution to implement fine-tuned perturbation searches. Inspired by humpback whales' coordinated hunting behavior, this process combines "surrounding prey" and "spiral attack" strategies to conduct multi-angle searches around the current optimal solution, aiming to uncover potentially better solutions within its local neighborhood. The algorithm employs update mechanisms such as D = |C·X* -X| and X(t+1) = X* -A·D, demonstrating WOA's capability to actively approach and deeply explore the neighborhood of optimal individuals in the solution space. Furthermore, we introduced three unique search strategies specific to WOA in the implementation: optimal solution-centered linear envelope perturbations, spiral trajectory approximation, and mutation perturbations that jump to other individual positions. In each WOA iteration, the execution ratio of these three strategies is controlled by a random variable p, ensuring local search accuracy while maintaining the possibility of escaping local optima, thereby enhancing overall convergence stability. To further strengthen WOA's dominance in the convergence phase, the algorithm dynamically adjusts the synergy ratio between GA and WOA across generations, implementing a weight mechanism that gradually shifts toward WOA (linearly increasing from 30% to 70% over iterations) to achieve a natural transition from global exploration to local development. Through these mechanisms, WOA is systematically integrated into the local fine-tuning module of this model, not only preserving diversity in solution space perturbations but also significantly improving convergence accuracy in later optimization stages and the robustness of LSTM prediction performance.

Once again, we sincerely appreciate your thoughtful feedback, which has helped us enhance the quality and clarity of our manuscript.

---

## [Editor Report · Decision Letter 3]

31 Jul 2025

Stock Market Forecasting Research Based on GA-WOA-LSTM

PONE-D-24-58534R3

Dear Dr. Wang,

We’re pleased to inform you that your manuscript has been judged scientifically suitable for publication and will be formally accepted for publication once it meets all outstanding technical requirements.

Kind regards,

Andrew Lewis

Academic Editor

PLOS ONE
---

## [Editor Report · Acceptance letter]

PONE-D-24-58534R3

PLOS ONE

Dear Dr. Wang,

I'm pleased to inform you that your manuscript has been deemed suitable for publication in PLOS ONE. Congratulations! Your manuscript is now being handed over to our production team.

Kind regards,

on behalf of

Dr. Andrew Lewis

Academic Editor

PLOS ONE